# Sudden Cardiac Death in the Young: State-of-the-Art Review in Molecular Autopsy

**Cecilia Salzillo, Vincenza Sansone and Francesco Napolitano ***

Department of Experimental Medicine, University of Campania "Luigi Vanvitelli", Via Luciano Armanni 5, 80138 Naples, Italy; cecilia.salzillo@unicampania.it (C.S.); vincenza.sansone@unicampania.it (V.S.)
* Correspondence: francesco.napolitano2@unicampania.it

**Abstract:** Sudden cardiac death (SCD) is defined as unexpected death due to a cardiac cause that occurs rapidly. Despite the identification of prevention strategies, SCD remains a serious public health problem worldwide, accounting for 15–20% of all deaths, and is therefore a challenge for modern medicine, especially when it affects young people. Sudden cardiac death in young people affects the population aged ≤ 35 years, including athletes and non-athletes, and it is due to various hereditary and non-hereditary causes. After an autopsy, if the cause remains unknown, it is called sudden unexplained death, often attributable to genetic causes. In these cases, molecular autopsy—post-mortem genetic testing—is essential to facilitate diagnostic and therapeutic pathways and/or the monitoring of family members of the cases. This review aims to elaborate on cardiac disorders marked by genetic mutations, necessitating the post-mortem genetic investigation of the deceased for an accurate diagnosis in order to facilitate informed genetic counseling and to implement preventive strategies for family members of the cases.

**Keywords:** sudden cardiac death; young; causes; molecular autopsy; molecular medicine

## 1. Introduction

Sudden death (SD) is defined as "a death occurs within one hour of the onset of symptoms, within 24 h of the last alive visit in the absence of witnesses, or for those resuscitated after cardiac arrest who die during hospitalization" [1]. SD is often the first clinical manifestation of a disease in apparently healthy subjects defined as asymptomatic, and an autopsy is the only instrument to define the cause of death [2].

Sudden cardiac death (SCD) is defined as "a natural and unexpected fatal event that occurs within one hour of the onset of symptoms, in an apparently healthy subject or in one in whom the illness was not so serious as to suggest a sudden event outcome" [3]. According to other authors, SCD is defined as "an unexpected and premature death caused by a cardiac condition in a person with known or unknown heart disease" [4].

Despite the identification of prevention strategies, death from cardiac causes is still the leading cause of death in the world; in particular, SCD represents a serious international public health problem and therefore a real challenge for modern medicine, especially when it affects young people.

Sudden cardiac death in the young (SCDY) is defined as "SCD that affects a population aged ≤ 35 years" [4–7], not athletes and athletes [8,9], with multiple hereditary/genetic or non-hereditary/genetic causes.

When the cause of death remains unknown despite an autopsy being performed, it is called sudden unexplained death (SUD), often caused by cardiomyopathies and cardiac channelopathies. In these cases, a post-mortem genetic study such as molecular autopsy (MA) is necessary.

In 2008 (Figure 1), the European Association of Cardiovascular Pathology (AECVP) published the "Guidelines for autopsy investigation of sudden cardiac death" with the

aim of standardizing the autopsy procedure in cases of SCD [10]. In 2017 (Figure 1), given the greater understanding of cardiovascular genetics, the introduction of new techniques, and the experience gained in the field, the following version was updated and published: "Guidelines for autopsy investigation of sudden cardiac death: 2017 update from the Association for European Cardiovascular Pathology" [2].

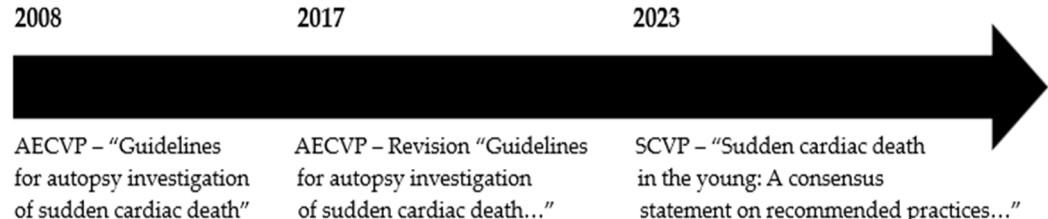

**Figure 1.** Temporal evolution of the guidelines and recommendations in SCD.

In 2023 (Figure 1), the Society for Cardiovascular Pathology (SCVP) published the recommendations "Sudden cardiac death in the young: a consensus statement on recommended practices for cardiac examination by pathologists from the Society for Cardiovascular Pathology" [4], underlining the importance of determining a genetic disease in a proband post-mortem to start a life-saving diagnostic–therapeutic and/or monitoring process for family members.

## 2. Sudden Cardiac Death in the Young

### 2.1. Epidemiology

SCD is a major public health challenge, contributing to approximately 15–20% of all mortality cases due to natural causes [4].

Each year, approximately 350,000 cases of sudden and unexpected deaths are recorded in Europe, while the United States records a range between 300,000 and 400,000 cases [11]. Previous investigations in the United States indicate an incidence ranging from 250,000 to 400,000, equivalent to an overall rate of 1–2 per 1000 inhabitants per year [12]. The European Society of Cardiology (ESC) reports incidence rates of between 36 and 128 deaths per 100,000 individuals per year, while other reviews cite figures ranging from 50 to 100 per 100,000 inhabitants [13].

The incidence of SCD among individuals aged $\leq 35$ years deserves particular attention and depends on the age considered. Indeed, in individuals aged < 30 years, the overall risk of MCI is approximately 1–2.8 per 100,000 individuals [14,15], with rates of 19% among children aged between 1 and 13 years and 30% among adolescents aged 14–21 years [16]. Alarmingly, in 33% of cases of sudden death, the cause of death remains unknown despite autopsy examination [17,18].

In Italy, the frequency of SCD and SCDY is based on ISTAT data due to the absence of Regional Registers and the National Register, with the exception of the Veneto region [3]. According to ISTAT data, SCD is responsible for approximately 50,000 deaths per year, while SCDY affects 1000 individuals every year. In particular, in the Veneto region, the cumulative incidence is equal to 1 per 100,000 per year, with rates of 0.9 per 100,000 per year among non-athletes and 2.3 per 100,000 per year among athletes [3].

### 2.2. Causes Hereditary Heart Diseases

The pathophysiological mechanisms underlying SCD can stem from mechanical origins or alterations in the conduction system. The causes of SCD are multifaceted and vary across different age groups.

In the elderly population, chronic structural diseases are predominant, with coronary artery disease accounting for 75% of SCD cases followed by cardiomyopathies, including dilated cardiomyopathy, hypertrophic cardiomyopathy, and arrhythmogenic right ventricular cardiomyopathy, which contribute to 15% of cases. Valvular heart diseases

represent 5% of cases, while genetic and acquired cardiac abnormalities and hereditary arrhythmic syndromes, such as long QT syndrome, Brugada syndrome, catecholaminergic polymorphic ventricular tachycardia, and early repolarization syndrome, constitute 3% and 2% of cases, respectively [6,19].

In individuals aged $\leq$ 35 years, inherited heart diseases are more prevalent. Hypertrophic cardiomyopathy accounts for 36% of cases, while arrhythmogenic right ventricular cardiomyopathy contributes to 20% [6,19]. Congenital heart disease is more frequently observed in neonates (less than 1 year of age).

Moreover, SCD can be categorized based on the certainty of the cause–effect relationship, namely, certain, highly probable, and uncertain; in such instances, a comprehensive assessment encompassing clinical history, circumstances of death, and ancillary tests such as toxicological, metabolic, and molecular studies become imperative [2].

Lastly, there exist cardiac conditions where delineating between physiological changes, secondary alterations, and pathological manifestations presents challenges, thus defining diagnostic gray areas [2,20].

### 2.3. Cardiomyopathy

Cardiomyopathies are a heterogeneous group of diseases characterized by the intrinsic pathology of the heart muscle associated with electrical and/or mechanical dysfunction and are among the most frequent causes of heart transplantation.

These conditions are classified based on morphology in hypertrophic cardiomyopathy, dilated cardiomyopathy, restrictive cardiomyopathy, arrhythmogenic cardiomyopathy, and non-compact cardiomyopathy and are classified by causes in primary and secondary cardiomyopathy or familial (genetic) and non-familial (non-genetic) cardiomyopathy.

Moreover, other classifications have been developed by the World Health Organization (WHO) in 1995, the American Heart Association (AHA) in 2006, the European Society of Cardiology (ESC) in 2008, and the phenotype–genotype–cause classification by the MOGE(S) in 2013 by the World Heart Federation (WHF).

Hypertrophic and arrhythmogenic cardiomyopathies are the most frequent cause of sudden cardiac death associated with sport, especially in young individuals.

### 2.3.1. Hypertrophic Cardiomyopathy (HCM)

Hypertrophic cardiomyopathy (HCM) is a cardiac condition characterized by significant primary hypertrophy of the left ventricle, with a wall thickness of 15 mm or greater. This hypertrophy leads to various functional impairments, including left ventricular outflow tract obstruction, diastolic dysfunction, mitral regurgitation, and myocardial ischemia.

While HCM can occur as a compensatory response or secondary to conditions such as hypertension or aortic stenosis, it typically presents with symptoms such as fatigue, dyspnea, chest pain, palpitations, and syncope. In some cases, it can lead to SCD due to malignant ventricular arrhythmias like rapid ventricular tachycardia and ventricular fibrillation [21].

The prevalence of HCM in the general population is approximately 0.2% [22], making it a relatively rare condition. However, it is the second most common cause of cardiomyopathy in the pediatric population, with a prevalence of 3 cases per 100,000 live births. The incidence of SCD in individuals with HCM ranges from 1% to 7.2%, with peak presentations occurring before 1 year of age and in the age group of 8–17 years [23]. According to some studies, it is the leading cause of SCD, while several studies have reported that it accounts for 2–36% of SCD cases [9].

Familial HCM often exhibits autosomal dominant inheritance, and more than 50 different pathogenic variants have been identified. Approximately 20–30% of cases are caused by pathogenic variants in genes encoding cardiac sarcomeric contractile proteins, including *MYH7*, *MYL2*, *MYL3*, *MYBPC3*, *TNNT2*, *TNNI3*, and *TPM1*. Pathogenic variants in *MYBPC3* and *MYH7* are particularly common, accounting for over 70% of cases (Figure 2) [22].

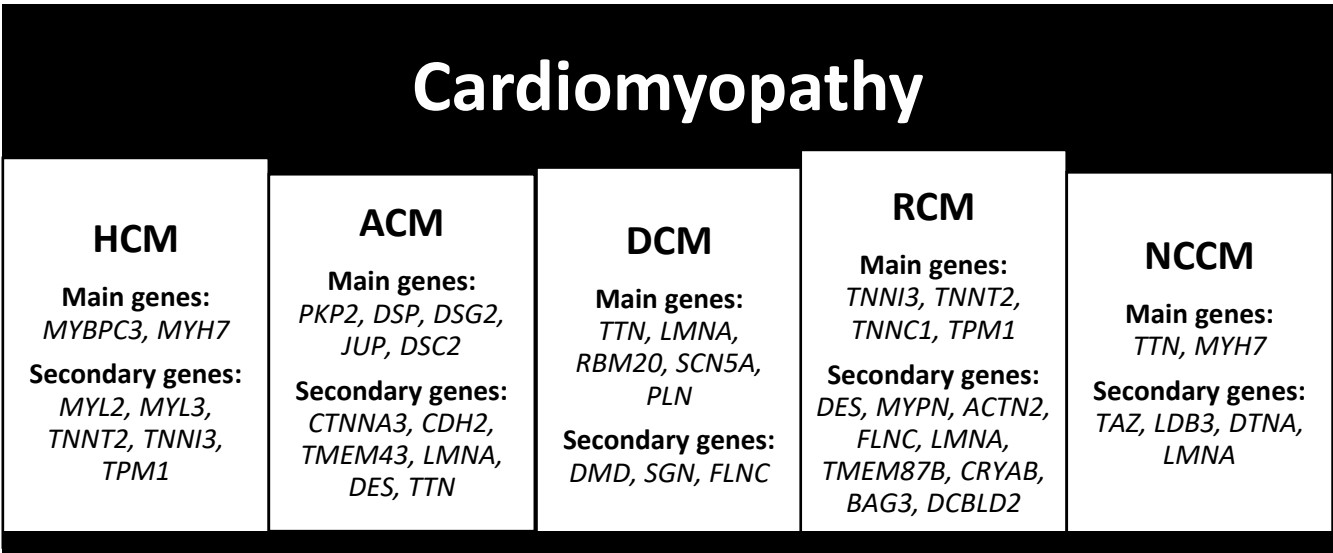

**Figure 2.** Main and secondary genes responsible for cardiomyopathies.

In addition to familial cases, other causes of HCM include mitochondrial gene pathogenic variants transmitted through the maternal line, referred to as mitochondrial hypertrophic cardiomyopathy [24], as well as inherited metabolic diseases, neuromuscular disorders, and RASopathies [23].

At autopsy, characteristic findings include a heart weighing over 500 g, predominant left ventricular hypertrophy, spiral myocardium, areas of whitish scars, and subaortic endocardial fibrosis plaque. Depending on the location of hypertrophy, HCM can be classified into four forms: hypertrophy of the anterior portion of the septum (type I), predominant septal hypertrophy (type II), hypertrophy of the interventricular septum and anterior wall (type III), and hypertrophy of all ventricular walls and the apex (type IV).

Histologically, HCM is characterized by hypertrophy of cardiomyocytes with hyperchromatic nuclei, the disorganization of cardiomyocytes and myofibrils, increased interstitial cellularity, the thickening of intramyocardial coronary arteries with disorganization of the media, intimal fibrosis, interstitial or replacement fibrosis, and focal myocarditis.

2.3.2. Arrhythmogenic Cardiomyopathy (ACM)

Arrhythmogenic cardiomyopathy (ACM), formerly known as arrhythmogenic right ventricular dysplasia/cardiomyopathy (ARVD), is a cardiac disorder characterized by the progressive fibrotic or fibroadipose replacement of the myocardium, leading to ventricular arrhythmias and sudden cardiac death, particularly among young athletes.

ACM has a prevalence ranging from 1:2000 to 1:5000, with a higher occurrence in males and clinical symptoms typically emerging between the second and fourth decades of life [25]. Among young athletes, the incidence of sudden cardiac death varies from 3% to 20% in some studies [25], while in individuals under the age of 65, it ranges from 5% to 10% [26].

In most cases, ACM is a genetic disorder with autosomal dominant inheritance, variable expressivity, and incomplete penetrance, resulting in diverse clinical presentations. The clinical spectrum includes progressive heart failure and ventricular arrhythmias such as ventricular tachycardia and ventricular fibrillation, often leading to sudden cardiac arrest as the initial manifestation.

Pathogenic variants in genes encoding structural proteins of cardiac intercellular junctions, particularly desmosomes, account for 30–50% of cases [27]. Specifically, the five genes encoding desmosomal proteins, in order of frequency, are plakophilin (*PKP2*) in 10–45%, desmoplakin (*DSP*) in 10–15%, desmoglein (*DSG2*) in 10%, and plakoglobin (*JUP*) and desmocolin (*DSC2*) in 1–2% [25]. Non-desmosomal genes involved in ACM

pathogenesis include alpha-T-catenin (*CTNNA3*), N-cadherin (*CDH2*), transmembrane protein 43 (*TMEM43*), lamin A/C (*LMNA*), desmin (*DES*), and titin (*TTN*), particularly in left or biventricular forms (Figure 2) [25].

In a minority of cases, ACM follows an autosomal recessive pattern, as observed in Naxos disease and Carvajal-Huerta syndrome, caused by homozygous mutations in *JUP* and certain *DSP* mutations, respectively [27].

The diagnosis of ACM is based on the Padua criteria [28], which consist of major and minor criteria varying according to the three phenotypic variants: classic or right dominant (ARVC), biventricular, and left dominant (ALVC). Notably, the diagnosis of ALVC without clinically demonstrable right ventricular anomalies necessitates the confirmation of a genetic mutation.

At autopsy, the heart typically exhibits increased weight, with a fibrotic or fibroadipose replacement of the myocardium, predominantly affecting the sub-epicardial ventricular region, either in the right, left, or biventricular distribution, depending on the phenotypic variants. Ventricular dilation with aneurysm formation, especially in the right ventricle, may also be observed.

Histologically, findings include fibro-adipose replacement, morphological alterations of cardiomyocytes with apoptosis and cytoplasmic vacuolization, and focal inflammatory infiltration.

### 2.3.3. Dilated Cardiomyopathy (DCM)

Dilated cardiomyopathy (DCM) is characterized by the dilation of the left and/or right ventricle, causing a progressive contractile dysfunction with a decrease in the cardiac ejection volume with each beat.

The residual volume produces an increase in the diameter and pressure of the ventricle; consequently, the compensatory response is hypertrophy and the progressive increase in the length of the sarcomeres to improve contractility. As a result, the heart gradually enlarges to a spherical shape.

DCM can be due to genetic and non-genetic causes including coronary artery disease, valvular disease, hypertension, myocarditis, alcoholism, drugs, and chemotherapy, and it can be associated with systemic diseases such as muscular dystrophy.

The prevalence of DCM is 1:250, of which 30–50% are familial cases [29] and approximately 40% of cases have an identifiable genetic cause [30], and DCM is responsible for 1–3% of SCD cases in athletes [9].

In most cases, DCM is autosomal dominant with variable expressivity and incomplete penetrance, but also specific forms of autosomal recessive, X-linked, mitochondrial inheritance, de novo mutations, polygenic disorders, and over 60 genes have been identified by genome sequencing.

In order of frequency, the protein coding genes are titin (*TTN*) in 20–25%, lamin A/C (*LMNA*) in 8%, RNA-binding protein 20 (*RBM20*) in 1–5%, type V cardiac voltage-dependent Na channel (*SCN5A*) in 2–3%, and phospholamban (*PLN*) (Figure 2) [30,31].

Furthermore, various genes encoding cardiac cytoskeletal proteins such as dystrophin (*DMD*) for Duchenne muscular dystrophy, sarcoglycan (*SGC*) for sarcolemmal instability muscular dystrophy, and filamin C (*FLNC*) are associated with arrhythmias and SCD (Figure 2) [31].

Genetic DCM is suspected in cases of family history with two first-degree relatives who have had idiopathic DCM or juvenile sudden cardiac death [31].

At autopsy, the heart weighs > 500 g and is globoid in shape, with the dilation of all cardiac chambers with the thinning of the walls and flattening of the trabeculae, as well as in areas of mural fibrosis and thrombosis.

The histology is characterized by degenerative changes and the variable size of the cardiomyocytes and with interstitial or replacement fibrosis.

2.3.4. Restrictive Cardiomyopathy (RCM)

Restrictive cardiomyopathy (RCM) is a rare cardiac muscle disorder characterized by a marked reduction in compliance, leading to disproportionately increased telediastolic ventricular pressure and impaired ventricular filling, often resulting in SCD in children.

RCM can manifest as either a primary familial or acquired condition or as a secondary complication of systemic diseases linked to congenital metabolic errors, infiltrative disorders, or skeletal myopathies [32]. The precise prevalence of RCM remains unknown, although its estimated incidence ranges from 0.03 to 0.04 per 100,000 children [32], with an average age of onset between 6 and 11 years [33]. Due to the rarity of primary RCM, the genetic underpinnings are not well understood. Conversely, secondary forms, such as amyloidosis linked to mutations in the transthyretin (*TTR*) gene or syndromic disorders like Alström syndrome or Myhre syndrome [34], are better characterized.

In most instances, RCM demonstrates autosomal dominant transmission, de novo mutations, or autosomal recessive inheritance patterns. Pathogenic variants have been identified in 19 genes encoding sarcomeric, cytoskeletal, or Z-disc proteins. Predominantly, pathogenic variants are concentrated within ten genes encoding sarcomere proteins, including cardiac troponins (*TNNI3*, *TNNT2*, and *TNNC1*) and alpha-tropomyosin (*TPM1*), with *TNNI3* being the most frequently implicated (Figure 2) [34]. Additionally, pathogenic variants have been identified in genes encoding non-sarcomeric proteins such as desmin (*DES*), myopalladin (*MYPN*), alpha-actinin-2 (*ACTN2*), filamin-C (*FLNC*), lamin A/C (*LMNA*), transmembrane protein 87B (*TMEM87B*), alphaB-crystallin (*CRYAB*), Bcl2-associated athanogene 3 (*BAG3*), discoidin, CUB, and LCCL domain-containing protein 2 (*DCBLD2*) (Figure 2) [34].

During autopsy, the heart typically appears normal or slightly diminished in size, with dilated atria and thickened, rigid walls exhibiting areas of fibrosis. Occasionally, mural thrombi may be present as a consequence of stasis.

Histologically, the findings may vary depending on the underlying causes but commonly include features indicative of cardiomyocyte degeneration and hypertrophy, areas of cellular disarray and interstitial fibrosis, the thickening of intramyocardial vessel walls, and focal inflammatory cell infiltration.

2.3.5. Non-Compaction Cardiomyopathy (NCCM)

Non-compaction cardiomyopathy (NCCM), also referred to as left ventricular non-compaction (LVNC) or hypertrabeculation, is distinguished by excessive trabeculation and profound intertrabecular recesses, presenting with a spectrum of clinical manifestations ranging from asymptomatic to heart failure, thromboembolism, arrhythmias, and SCD. It can involve the right, left, or both ventricles and may manifest as an isolated condition or in conjunction with other cardiomyopathies and severe congenital heart disease.

This condition is rare, and its prevalence in the general population remains unknown; however, some studies have reported an estimated prevalence ranging from 0.014% to 1.3% based on echocardiographic findings. Familial cases account for approximately 20–40% of occurrences, rendering NCCM the third most common cardiomyopathy in pediatric patients [35].

NCCM primarily exhibits autosomal dominant transmission, although autosomal recessive and X-linked inheritance patterns have also been documented. Pathogenic variants in genes encoding sarcomeric, cytoskeletal, and nuclear membrane proteins disrupt embryonic myocardial compaction, leading to NCCM.

The most prevalent genetic pathogenic variants involve sarcomeric proteins, such as *TTN* (including *ACTC* and *SNTA1*) and *MYH7* (including *SCN5A* and *NKX2-5*), as well as cytoskeletal protein mutations like *TAZ* (associated with Barth syndrome, including *TNN13* and *P121L*), *LDB3* mutations (including *TNNT2* and *PRDM16*), *DTNA* (including *TNNT2* and *TPM1*), and *LMNA* (including *MYBPC3* and *HCN4*) (Figure 2) [35,36].

Macroscopically, NCCM is typified by an abundance of non-compacted myocardium relative to normal myocardium (with a ratio > 2) at the apical and mid-third levels, featuring polypoid or anastomotic trabeculae.

Histologically, deep intertrabecular recesses extending from the endocardium to the epicardium are observed, with depths exceeding 50% of the thickness of the left ventricular wall.

### 2.4. Cardiac Channelopathies

Cardiac channelopathies are a group of inherited heart diseases caused by abnormalities in the genes that code for cardiac regulatory proteins, sodium channels, potassium channels, calcium channels, and adrenergic receptors.

These anomalies affect the cardiac action potential or intracellular calcium homeostasis, with electrical instability and a greater predisposition to malignant arrhythmias.

The most frequent channelopathies responsible for SCDY are long QT syndrome (LQTS), Brugada syndrome (BrS), short QT syndrome (SQTS), catecholaminergic polymorphic ventricular tachycardia (CPVT), and early repolarization syndrome (ERS).

They predominantly have an autosomal dominant transmission with incomplete penetrance and variable expressivity, so the clinical manifestation varies from asymptomatic to malignant arrhythmias.

Mutations in sodium channels such as *SCN5A*, *SCN4B*, and *SCN5A* alter the phase of depolarization and manifest as LQTS and BrS. Mutations in potassium channels such as *KCNQ1*, *KCNH2*, *KCNE1*, *KCNE2*, *KCNJ2*, and *KCNJ5* alter the repolarization phase and manifest as LQTS and SQTS. Pathogenic variants affecting components of voltage-gated calcium channels such as *CACNA1C* and calmodulin and its subtypes such as *CALM1*, *CALM2*, *CALM3*, and caveolin-3 can cause arrhythmias, LQTS, and Timothy syndrome [37].

They are characterized by the absence of structural heart disease, and the first manifestation can be SCD; in these cases, the only way to reach a diagnosis is via a molecular autopsy which is essential to the direct genetic counseling of family members and identifying carriers of genetic pathogenic variants.

Genetic screening can often give negative results in 20% of patients with LQTS and BrS, in 80% with SQTS, and in 40% with CPVT [37].

### 2.4.1. Long QT Syndrome (LQTS)

Long QT syndrome (LQTS) is a cardiac channelopathy characterized by delayed myocardial repolarization, causing prolonged QT intervals on the electrocardiogram (ECG) and clinically manifesting with ventricular tachyarrhythmias such as torsades de pointes (TdP) with an increased risk of MCIG.

LQTS can be congenital or acquired and are associated with pharmacological or electrolyte imbalance.

The prevalence ranges approximately from 1 in 2000 individuals to 1 in 2500 individuals [38], and patients with positive genetics but asymptomatic have a 36% probability of a non-fatal cardiac event and a risk of MCI of 13%, approximately 50% of survivors have a probability of recurrence of ventricular fibrillation [39], and it is a major cause of SCD from ventricular arrhythmias in young people and athletes [40].

Moreover, 17 genes have been identified as responsible of LQTS, with 3 major genes such as *KCNQ1* (LQT1), *KCNH2* (LQT2), and *SCN5A* (LQT3) responsible for 75% of cases and 14 minor genes such as *AKAP9*, *CACNA1C*, *CALM1*, *CALM2*, *CALM3*, *CAV3*, *KCNE1*, *KCNE2*, *KCNJ2*, *KCNJ5*, *SCN4B*, *SCN5A*, *SNTA1*, and *TRDN* responsible for less than 5% of cases (Figure 3) [41].

Moreover, there are three atypical LQTS or multisystem syndromic disorders including Ankyrin-B syndrome gene *ANK2* (LQT4), Anderson–Tawil syndrome (ATS) gene *KCNJ2* (LQT7), and Timothy syndrome (TS) gene *CACNA1C* (LQT8) [42].

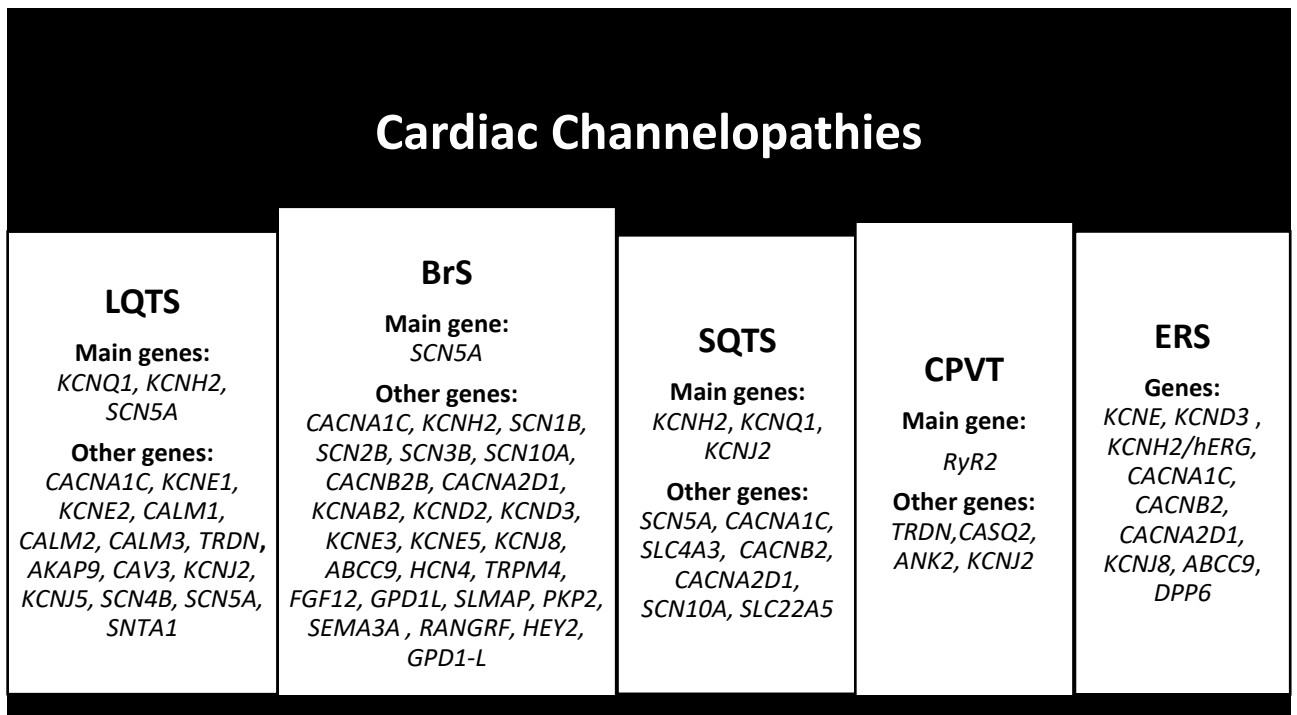

**Figure 3.** Main and other genes associated with cardiac channelopathies.

2.4.2. Brugada Syndrome (BrS)

Brugada syndrome (BrS) is an autosomal dominant hereditary cardiac channelopathy characterized by anomalies in repolarization, depolarization, and current–load correspondence, with accentuated J wave and coved-type ST segment elevation in the right precordial leads V1–V3 on an ECG, responsible for tachycardia polymorphic ventricular fibrillation and ventricular fibrillation and with an increased risk of MCIG.

Three types of ECGs have been described: type 1 ECG is characterized by an ST segment elevation of ≥2 mm in V1–V3, followed by negative T waves, type 2 ECG has >2 mm of ST segment elevation and a saddle-shaped ST segment, and type 3 has the morphology of type 1 or type 2, but with <2 mm of ST segment elevation [43].

It is responsible for 4% of all SCD cases and almost 20% of SCD in individuals with structurally normal hearts [44], and in fact, 80% of patients are asymptomatic until MI occurs [45].

The global prevalence of BrS is estimated to be around 0.05%, and in Southeast Asia, it represents the main cause of natural death in males under the age of 50 [46] with a prevalence 14 times higher than that worldwide.

It is generally asymptomatic, whereas the symptoms usually occur following triggering events such as feverish episodes, large meals, or during sleep. Syncope is the most frequent symptom, and other clinical manifestations are sudden infant death syndrome (SIDS) and sudden and unexpected nocturnal death syndrome (SUNDS).

BrS is associated with over 300 *SCN5A* pathogenic variants with incomplete penetrance and variable expression, representing 11–28% of cases, and pathogenic variants in other genes have been detected such as *SCN1B, SCN2B, SCN3B, SCN10A, CACNA1C, CACNB2B, CACNA2D1, KCNAB2, KCND2, KCND3, KCNE3, KCNE5, KCNJ8, KCNH2, ABCC9, HCN4, TRPM4, FGF12, GPD1L, SLMAP, PKP2, SEMA3A, RANGRF, HEY2*, and *GPD1-L* (Figure 3) [46].

### 2.4.3. Short QT Syndrome (SQTS)

Short QT syndrome (SQTS) is a rare autosomal dominant inherited cardiac channelopathy characterized by a shortened QTc interval (<340 ms), and it is associated with tachyarrhythmias and an increased risk of SCDY.

The most frequent clinical presentation of SQTS is SCD (30% of cases), and it is often the first clinical manifestation of the disease [47], especially in the first year of life and among individuals aged between 20 and 40 years. It is difficult to accurately estimate the prevalence of SQTS due to the rarity of occurrence and the diagnosis of this pathological entity [48].

SQTS is caused by mutations in potassium and calcium channel genes, and six genes responsible for the different subtypes have been identified which explain only 25% of cases. Specifically, the SQT1, SQT2, and SQT3 subtypes are caused by gain-of-function mutations in the *KCNH2*, *KCNQ1*, and *KCNJ2* genes, respectively. Conversely, the SQT4, SQT5, and SQT6 subtypes are caused by loss-of-function mutations in the *CACNA1C*, *CACNB2*, and *CACNA2D1* genes, respectively [37]. Other mutations are in the *SCN5A*, *SLC4A3* and *SCN10A* genes, respectively, the SQT7, SQT8, and SQT9 subtypes, and finally mutations in the *SLC22A5* gene (Figure 3) [49].

### 2.4.4. Catecholaminergic Polymorphic Ventricular Tachycardia (CPVT)

Catecholaminergic polymorphic ventricular tachycardia (CPVT) is a hereditary cardiac channelopathy caused by mutations in ion channels characterized by the presence of polymorphic VT and bidirectional VT, triggered by physical or emotional stress and manifesting with malignant arrhythmias and MCIG.

It has an incidence of approximately 1 in 5000/10,000 individuals, with a mortality rate of up to 30–35% within 30 years of age in untreated patients [50] and up to 13% in patients receiving therapy, and 30% have a family history of MCI before the age of 40 [49].

The most frequent symptom is syncope, and its presentation varies from two years to 21 years of age, with a poor prognosis of up to 50% mortality in those who are 20 years of age [50].

CPVT has been classified based on the gene pathogenic variant into five subtypes: CPVT1, which involves the *RyR2* gene with autosomal dominant transmission, is responsible for approximately 60% of cases; CPVT3, which involves the *TRDN* gene with autosomal recessive transmission, manifests around the age of 10 age; CPVT2 involving the *CASQ2* gene with autosomal recessive transmission manifests around seven years of age; and CPVT4 involving the *ANK2* gene and CPVT5 involving the *KCNJ2* gene are autosomal dominant and manifest at 4 and 2.5 years of age and with a frequency of <1% (Figure 3) [50,51].

### 2.4.5. Early Repolarization Syndrome (ERS)

Early repolarization syndrome (ERS) is an inherited channelopathy, and it is a variant of J wave syndromes like SB with which it shares multiple similarities. The syndrome is defined as the occurrence of an early repolarization (ERP) pattern in individuals who have been successfully resuscitated from a confirmed case of unexplained VF or polymorphic VT [52], with an increased risk of malignant arrhythmias and SCD.

It has a prevalence in the general population between 1.3% and 13.3%, with higher prevalence in physically active individuals up to 33.9%, and in men [48,53], and it is responsible for 7.3% of SCD [48].

ERS is classified into three different subtypes based on the spatial localization of the early repolarization (ER) pattern in the lateral (type 1), infero-lateral (type 2), or right infero-lateral and precordial (type 3) leads [37].

It is caused by missense mutations in ion channels, specifically pathogenic variants in the *KCNE*, *KCND3*, and *KCNH2/hERG* genes for the VGKC channel; pathogenic variants in the *CACNA1C*, *CACNB2*, and *CACNA2D1* genes for the VGCC channel; and pathogenic variants in the *KCNJ8*, *ABCC9*, and *DPP6* genes for the Okay channel (Figure 3) [37,48].

### 3. Molecular Autopsy (MA)

The term "autopsy" originates from Greek roots, combining "Auto" (αὐτό), meaning "self" or "from oneself," and "Opsis" (ὄψις), meaning "vision" or "observation." Thus, it translates to "to observe from oneself" or "to see from oneself".

Autopsy can be categorized into two main types: a hospital autopsy and a forensic autopsy. A hospital autopsy involves the examination of a deceased individual by a pathologist to determine the cause of death and to validate or revise diagnoses made during the person's lifetime on behalf of the health authority. On the other hand, a forensic autopsy is conducted by a forensic pathologist to establish the time, cause, means, and manner of death, especially in cases where foul play is suspected, on behalf of the judicial system.

Autopsy is the only tool to definitively determine the cause of death post-mortem, and according to some studies, in 5–10% of cases, the cause remains unknown, termed SUD [54,55]. Often, in these instances, the cause of SCDY is a malignant hereditary arrhythmogenic disease (MHAD) [54,55].

MHAD can be caused by cardiomyopathies characterized by a structurally altered heart, pathogenic variants in genes encoding contractile sarcomeric proteins, and pathogenic variants in genes encoding ion channels (sodium, chloride, and potassium).

MHAD predominantly exhibits autosomal dominant inheritance, with incomplete penetrance, variable expressivity, genetic overlap, and pleiotropy, leading to variable clinical manifestations and consequently challenges in genetic diagnosis and risk stratification, especially in asymptomatic carriers [56,57].

Furthermore, SCD has been associated with oligogenic models of heart disease, where rare potentially pathogenic variants may synergistically contribute to the risk of SD, and an over-representation of rare variants in cardiac genes has been identified in young deceased individuals [58,59].

Moreover, more common variants might act as modulatory factors to rare variants causing a more or less severe phenotype, and variants in non-coding regions (intronic regions and UTRs) or regions encoding microRNAs can modulate or play a key role [60].

MHAD predispose individuals to alterations in electrical conduction, leading to malignant arrhythmias and SCD. Notably, nearly 30% of SCD cases in the young population remain without a conclusive cause of death after a comprehensive autopsy examination [56]. In these instances, MA, which is a post-mortem genetic analysis, becomes crucial to enable diagnoses in the decedent and guide the family towards genetic counseling.

A post-mortem genetic analysis is conducted on 5–10 milliliters of blood preserved with ethylenediaminetetraacetic acid (EDTA) or on tissues either frozen or fixed in formalin and embedded in paraffin (FFPE) collected during the autopsy [4]. This analysis identifies a genetic mutation in up to 25% of cases and, according to other studies, in 10–20% of SCDY cases [55].

Cardiac channelopathies are the most common cause of SUD in the pediatric and young population (Table 1).

LQTS, depending on the genetic mutation, is associated with physical exercise, swimming, and sleep; in 85% of cases, the genetic analysis of all genes identifies the cause of the disease, with 80% carrying the pathogenic variant in three principal genes, *KCNQ1*, *KCNH2*, and *SCN5A*, which are recommended by current guidelines [52,61].

BrS should be suspected in young individuals who died suddenly at night; genetic analysis identifies causative gene mutations in 35% of cases, with 30% of patients carrying the pathogenic variant in *SCN5A*, the only one recommended by current guidelines to be analyzed [52,61].

SQTS should be suspected in a newborn who died suddenly, being considered the main cause of death in the first year of life (sudden infant death syndrome—SIDS); genetic analysis identifies pathogenic variants primarily in three genes, *KCNQ1*, *KCNJ2*, and *KCNH2*, in 45% of cases, as recommended by current guidelines [52,61].

**Table 1.** Malignant hereditary arrhythmogenic diseases—cardiac channelopathies and recommended gene mutation for diagnosis.

| Cardiac Channelopathies | Recommended Gene Mutations | Suspected |
|---|---|---|
| LQTS | *KCNQ1, KCNH2, SCN5A* (80%) | Young died suddenly during physical exercise or swimming or sleep |
| BrS | *SCN5A* (30%) | Young died suddenly at night |
| SQTS | *KCNQ1, KCNJ2, KCNH2* (45%) | Newborn died suddenly |
| CPVT | *RyR2* (55%) | Young died suddenly during adrenergic stress. |

CPVT should be suspected in young patients who died suddenly during adrenergic stress. It is one of the leading causes of SCD, especially in adolescents and children before the age of 10. Genetic analysis identifies mutations in 65% of cases, with 55% due to genetic alterations in the *RyR2* gene, as recommended by current guidelines [52,61].

Cardiomyopathies are the most frequent cause of SUD in the population aged < 40 years (Table 2).

**Table 2.** Malignant hereditary arrhythmogenic diseases—cardiomyopathies.

| Cardiomyopathies | Most Frequent Gene Mutations |
|---|---|
| HCM | *MYBPC3* (40–45%), *MYH7* (15–25%) |
| ACM | *PKP2* (10–45%), *DSP* (10–15%), *DSG2* (10%), *JUP* and *DSC2* (1–2%) |
| DCM | *TTN* (20–25%), *LMNA* (8%), *RBM20* (1–5%), *SCN5A* (2–3%) |

Even when no structural abnormalities are identified post autopsy, it is imperative to conduct a molecular autopsy of the associated genes. In cases where a pathogenic variant is identified in the associated genes, further investigations are needed to conclude whether the arrhythmia occurred before the structural alteration. Indeed, a concealed cardiomyopathy may occur whereby the arrhythmia manifests before the structural alteration [62,63]. Furthermore, detailed diagnostics are needed to exclude other conditions with overlapping phenotypes.

The genes most frequently mutated in HCM are *MYBPC3* and *MYH7*, accounting for 70% of cases, in particular *MYBPC3* for 40–45% and *MYH7* for 15–25% [22,55].

The most commonly mutated genes in ACM are *PKP2* in 10–45% of cases, *DSP* in 10–15%, *DSG2* in 10%, and *JUP* and *DSC2* in 1–2% [25].

The genes most frequently mutated in DCM are *TTN* in 20–25% of cases, *LMNA* in 8%, *RBM20* in 1–5%, and *SCN5A* in 2–3% [30,31].

Despite MA being recommended in cases of SUD suspected to be caused by MHAD [4], it is not included in the autopsy protocols of both hospital and forensic settings in most countries. Recent data show that only 37% of SUD cases with a suspected arrhythmic cause undergo post-mortem genetic testing [64,65].

Furthermore, due to the increasing number of rare variants that remain ambiguous—defined variants of unknown significance (VUS) [66–68]—a significant portion of SUD cases concludes without definitive answers after molecular autopsy. In order to reduce the frequency of ambiguous variants, it has been reported that rare variants associated with hereditary arrhythmogenic syndromes should be reanalyzed and reclassified within five years [68–70] if already classified according to the American College of

Medical Genetics and Genomics and the Association for Molecular Pathology recommendations [71].

Therefore, it would be beneficial to develop guidelines or recommendations focused on the interpretation of genetic variants in the context of autopsy procedures.

### 4. Conclusions and Future Prospects

The identification of a genetic–molecular alteration underlying a malignant hereditary arrhythmic syndrome is essential to guide genetic counseling and to implement personalized preventive strategies such as the monitoring and/or therapy of the patients' family members in order to reduce sudden cardiac death.

The challenge for modern medicine is to ensure that molecular autopsy becomes routine in autopsy practice to ensure the correct genetic interpretation of identified variants and their clinically useful translation into asymptomatic patients. Therefore, for the appropriate management of these cardiac syndromes, it is essential to enable the formation and collaboration of a multidisciplinary team composed of a cardiovascular pathologist, a forensic pathologist, a geneticist, a cardiologist, a pediatrician, and a general practitioner.

**Author Contributions:** C.S. and F.N. participated in the conception and design of the study; C.S., V.S. and F.N. contributed to the data analysis and interpretation; C.S. and F.N. wrote the article. All authors have read and agreed to the published version of the manuscript.

**Funding:** This research received no external funding.

**Conflicts of Interest:** The authors declare no conflicts of interest.

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
