# Peer review of "Sudden Cardiac Death in the Young: State-of-the-Art Review in Molecular Autopsy"

_cimb, doi:10.3390/cimb46040207_

Round 1

Reviewer 1 Report

Comments and Suggestions for Authors

The reviewer thanks the authors for their comprehensive review article on one of the most captivating topics within cardiac electrophysiology, the unprovoked death of the young and seemingly healthy. Though the article has merit, a few central concerns remain that prohibit this reviewer from completing the entire peer review:

1. The novelty of this article is not clear given numerous articles have been published on this exact content matter. Specifically in reference to:

a) Campuzano O, Sarquella-Brugada G. Molecular autopsy in sudden cardiac death. Glob Cardiol Sci Pract. 2023 Jan 30;2023(1):e202308. doi: 10.21542/gcsp.2023.8. PMID: 36890841; PMCID: PMC9988296.

b) Semsarian C, Ingles J, Wilde AA. Sudden cardiac death in the young: the molecular autopsy and a practical approach to surviving relatives. Eur Heart J. 2015 Jun 1;36(21):1290-6. doi: 10.1093/eurheartj/ehv063. Epub 2015 Mar 11. PMID: 25765769.

c) Martínez-Barrios E, Grassi S, Brión M, Toro R, Cesar S, Cruzalegui J, Coll M, Alcalde M, Brugada R, Greco A, Ortega-Sánchez ML, Barberia E, Oliva A, Sarquella-Brugada G, Campuzano O. Molecular autopsy: Twenty years of post-mortem diagnosis in sudden cardiac death. Front Med (Lausanne). 2023 Feb 10;10:1118585. doi: 10.3389/fmed.2023.1118585. PMID: 36844202; PMCID: PMC9950119.

2. The article would benefit from figure(s) to assist in illustration of the main/central concepts.

Comments on the Quality of English Language

The article contains few minor grammatical errors.

Author Response

Reviewer 1

  1. In response regarding the novelty of the article, the novelty of this review lies in addressing Sudden Cardiac Death in the Young (SCDY) as a state of the art (past, present, future) concerning the suggested references that have been cited in this paper (references 55 and 56). Indeed, the manuscript described the evolution of the Sudden Cardiac Death guidelines up to the recommendations of the SCDY of the Society for Cardiovascular Pathology that, only in 2023, addresses the topic in young people, in light of the identification of the new gene mutations. Moreover, the review focuses on cardiomyopathies and cardiac channelopathies with a pathologist's approach, highlighting in cardiomyopathies, in addition to genetics, also the macroscopic and histological characteristics when present, and in channelopathies the absence of macroscopic and histological alterations. The review addresses also the importance of genetic diagnosis on the victim and family members with a new vision of the autopsy as a prevention tool and, therefore, as a future proposal for the creation of an international standardized protocol.

My colleagues and I thank the reviewer for pointing out the article "Semsarian C, Ingles J, Wilde AA. Sudden cardiac death in the young: the molecular autopsy and a practical approach to surviving relatives. Eur Heart J. 2015;36(21): 1290-1296" that has been added in the references section (reference 67) due to the addition of the paragraph on variants of unknown significance (VUS).

  1. As suggested, throughout the manuscript we have included three explanatory figures to assist the readers in the illustration of the main concepts. In particular, Figure 1 (line 60) explains the temporal evolution of the 2008 and 2017 SCD guidelines up to the 2023 SCDY recommendations; Figure 2 (line 293) represents the main and secondary genes of cardiomyopathies; Figure 3 (line 419) illustrates the main and strongly associated genes of cardiac channelopathies.

Reviewer 2 Report

Comments and Suggestions for Authors

Sudden Cardiac Death in the Young: State-of-the-Art Review in Molecular Autopsy reviews molecular profiling for genetic causes of sudden coronary death. In fact, this is indeed a very pressing topic, since almost 90% of such deaths remain causeless. Especially in developing countries where there is no adequate coverage of genetic screening due to poverty or some other reasons. And, most importantly, sudden coronary death is, for the most part, the death of young men. This is a very important, complete and up-to-date work by the authors. They are great guys because they did a very good job. The review covers the problem so completely and well that the reviewer enjoyed getting to know it. Indeed, the work takes into account both epidemiology and genetics, and there is information about rare genetic syndromes and the causes of prolongation of the Q T interval. The review is worthy of being published in a scientific journal.

Author Response

My colleagues and I are most grateful for the extremely positive tone of the reviewer’s comments.

Reviewer 3 Report

Comments and Suggestions for Authors

Authors written an interesting revision about molecular autopsy in the area of inherited cardiomyopathies/channelopathies. The revision include last data about recent studies in the field. Some point should be clarified:

1.- please genes should be written in italic.

2.- please, modify the term "mutation". It refers to a wide spectrum of alterations in the genome. It should be used "rare variant" or "pathogenic/likely pathogenic variant" if applies.

3.- Please check all genes due to some mistakes: ex, line324 KCNQ should be KCNQ1.

4.- Table 1, SrB should be BrS. Please check also in the text.

5.- What to do if a variant remains with an ambigous significance, called VUS? Actionable? Reclassification?

6.- Concerning all diseases mentioned in the text, what about genes considered definite/strong instead minor genes? Could be interesting to divide genes if main or minor as well as with a definite disease-association or just suspected.

Author Response

  1. As suggested, we have written the genes in italics throughout the manuscript.
  2. As suggested, we have changed the term “mutation” to “pathogenic variant” if applicable throughout the manuscript.
  3. As suggested, we have corrected the KCNQ gene to KCNQ1 (line 332) and checked the other genes throughout the text.
  4. As suggested, in the Table 1 we have replaced SrB with BrS and it has been verified in the text.
  5. As suggested, in the Molecular Autopsy (MA) section we have added a new paragraph (lines 505-510) and relative references (references 67-70) regarding the classification of the variants of unknown significance (VUS).
  6. As suggested, in the Molecular Autopsy (MA) section we have added a paragraph (lines 477-480) regarding the gene types. Moreover, in Figure 3 (line 419) we have divided the main genes and genes strongly associated with cardiac channelopathies and in Figure 2 we have divided into main and secondary genes (line 293).

Round 2

Reviewer 1 Report

Comments and Suggestions for Authors

Thank you

Author Response

My colleagues and I are most grateful for the reviewer’s thanks.

Reviewer 3 Report

Comments and Suggestions for Authors

Authors modified the manuscript accordingly to recommendations.

Some points to modify:

1.- Figure 3 concerning main genes associated with BrS (as well as in the text concerning the same issue). Only SCN5A is the main gene associated with BrS. All other genes have been associated with similar phenotypes. Actually, CACNA1C is associated with a BrS-like with shorter than normal QT... not BrS. In contrast, in table 1, only SCN5A is mentioned... please clarify.

2.- Line 507-510... ACMG recommend reclassification? Please clarify this point. The ACMG guidelines published in 2015 does not mention any recommendation concerning reclassification.

3.- Concerning italic format of genes, only the abbreviation should be written in italic... ex: the titin gene (TTN)... or the desmin gene (DES)...

Author Response

  1. As suggested, in the Figure 3 we have clarified the main and other genes associated with cardiac channelopathies and in the Table 1 we have replaced “Most frequent gene mutations” with “Recommend gene mutations” in order to avoid misunderstanding. Moreover, we have changed the titles of Table 1 and Figure 3.
  2. As suggested, in the Molecular Autopsy section we have modified the mistake related to ACMG (lines 507-514) and we have added the references of previous studies that recommend the reclassification of variants every 5 years: [69] Martinez-Barrios, E.; Sarquella-Brugada, G.; Perez-Serra, A.; et al. Reevaluation of ambiguous genetic variants in sudden unexplained deaths of a young cohort. Int J Legal Med. 2023, 137(2), 345-351; [70] Campuzano, O.; Sarquella-Brugada, G.; Fernandez-Falgueras, A.; et al. Reanalysis and reclassification of rare genetic variants associated with inherited arrhythmogenic syndromes. 2020, 54, 102732; [71] Sarquella-Brugada G.; Fernandez-Falgueras, A.; Cesar, S.; et al. Clinical impact of rare variants associated with inherited channelopathies: a 5-year update. Hum Genet. 2022, 141(10), 1579-1589.
  3. As suggested, we have written only the abbreviation of genes in italics throughout the manuscript.

Round 3

Reviewer 3 Report

Comments and Suggestions for Authors

No comments